# Application of Conjoint Analysis and 5Ps Marketing Mix in Identifying Customer Preference of Alcohol Consumption among Youths in the Philippines

**DOI:** 10.3390/foods12071401

**Published:** 2023-03-26

**Authors:** Jenalyn Shigella G. Yandug, France D. Ponce, Ardvin Kester S. Ong

**Affiliations:** School of Industrial Engineering and Engineering Management, Mapúa University, 658 Muralla St., Intramuros, Manila 1002, Philippines

**Keywords:** alcoholic beverages, conjoint analysis, consumer preferences, flavors

## Abstract

Alcoholic beverages have long been one of the most popular commodities. They have been mass-produced worldwide because of their popularity and demand. On a survey by Statista in 2021, alcohol consumption is projected to increase in the Philippines. Thus, the popularity of numerous alcoholic beverages piqued the curiosity of consumers and researchers alike. This study used conjoint analysis that aims to have a better understanding of consumer preferences for alcohol consumption among youths. Additionally, the study considered the following attributes: (1) type of alcoholic beverage, (2) oral sensation, (3) flavor, (4) origin, (5) color, and (6) price. Results revealed that the product’s price is the most crucial attribute influencing consumer preferences by 26.311%, followed by the type of alcoholic beverage with an importance score of 26.237%. The least considered attribute is the product’s color, having an importance score of 7.790%. These were supported by two statistical tests, the Pearson’s Correlation Test and Kendall’s Tau Test, which both obtained a value higher than 0.8. Managerial implications were presented in the study to help provide strategies and development for alcoholic beverage distribution in the Philippines based on the findings of this study, which relate to young consumers.

## 1. Introduction

Alcoholic beverages are one of the most popular consumed products. They are massively produced worldwide due to their popularity and demand in different countries. In fact, according to the World Health Organization (WHO) [1], most eastern European countries have the highest alcohol consumption in the world. In addition, the consumption of alcohol in 2005 was equal to 6.13 L of pure alcohol consumed per person at the age of at least 15 years old. It was added that 45% of the recorded alcohol is consumed in South East Asia, mainly in the form of Spirit Alcohols [1]. The popularity of different alcoholic beverages not only attracts consumers but has been the subject of various research endeavors.

The consumption of alcoholic beverages began in medieval times. In the 18th century, alcoholic beverages such as wine were consumed for different reasons [2]. One of the reasons was because of the religious belief that it purified one’s soul. This was mainly performed during this time in Greece [2]. The popularity of alcoholic products over the decades also attracted many researchers to study their effects on human health. According to Hamdan-Mansour [2], despite the immense consumption of alcohol over the past centuries, little to no knowledge about its influence on our bodies exists. However, throughout the years, medical studies have been conducted to assess the impacts of alcoholic beverages on the human body. According to Murphy et al. [3], the consumption of alcohol may increase the likelihood of developing diseases or disorders such as breast cancer, colorectal cancer, and dementia.

Moreover, Gronbaek [4] mentioned in his study that a high intake of alcohol causes cancer. In addition, he also mentioned that people who consume too much alcohol would most likely have a poor nutritional status. In contrast, Gronbaek [4] also argued that alcohol consumption has beneficial effects proven by epidemiological evidence. According to his research, the right amount of alcohol intake, typically categorized as light to moderate drinkers, is less likely to cause cardiovascular disease, thus lowering the risk of death. The same findings are discovered by Corrao et al. [5], who found that a light to moderate intake of alcohol provides a cardio-protective effect that lowers risk levels even for different genders.

Several studies have been conducted on the impacts of alcohol consumption on psychological conditions. According to Vinder-Caerols et al. [6], it is believed that a high intake of alcoholic beverages may alter behavior either temporarily or permanently. On the other hand, a study conducted by Cheng et al. [7] stated that higher stress and anxiety levels could increase alcohol cravings. The studies also showed that controlling specific predictors may cause an insignificant relationship between alcohol cravings and depression; this is in support of the study by Vinader-Caerols [6], that temporary alteration in behavior may exist once a person consumes alcohol but can still recover from the said addiction. Similar findings were found in the study of Irizar et al. [8], wherein people who are already drinking at hazardous levels will most likely increase their consumption more than those who are drinking to cope with stress, depression, and anxiety. Subsequently, Rodriguez-Miguez and Nogueira [9] also argued that alcohol consumption leads to disorders that influence intangible adverse effects such as deterioration of social and family relationships. In contrast, a study written by Labajo [10] mentioned that in the Philippines, alcoholic beverages were consumed as a tool to pursue socializing or bonding with different people. Thus, despite its negative impacts, there are positive impacts from drinking alcohol. With the benefits and disadvantages brought by alcoholic beverages, high consumptions are still evident.

Despite the effects of alcohol on both physical and psychological health, a significant number of consumers still pursue drinking for different reasons. In the United States, a study was conducted by Capasso et al. [11] to determine whether there has been an increase or decrease in alcohol consumption due to the COVID-19 pandemic. The study revealed that the pandemic severely affected mental health, which led to a 29% increase in alcohol consumption. Additionally, more women drank alcohol at 33% compared to males at 24% during the pandemic. Likewise, in Metro Manila, Philippines, a survey was conducted by Pagkatipunan [12] to determine the number of current college drinkers in the said year. The survey lasted 30 days, and based on its results, 30.2% of its population are current drinkers. Results showed that people are inclined to participate in drinking activities at a young age despite its impacts on human and psychological health.

According to Calina et al. [13], various studies have projected increased alcohol consumption during the pandemic. Their study mentioned a 146% increase in alcohol consumption in eastern Europe. An increase in alcohol consumption is also evident in Greece for many reasons, such as coping with anxiety and depression. Likewise, increased alcohol and tobacco consumption was proven to exist in Belgium during the pandemic. The Philippines also projected that alcohol consumption would increase throughout the year since results indicated that there would be an increase in revenue due to alcohol sales, based on a survey by Statista [14].

Despite the influence mentioned above brought about by alcohol, it is undeniably a fact that there is still an increasing number of customers that devour alcoholic beverages. With that, different manufacturing companies started producing alcoholic beverages of various types, flavors, and appearances to provide customers with other options. With this, there is high competition within the alcohol industry or market. 

Focusing on the Philippines, a limited number of studies focusing on alcoholic beverage and their consumption is seen. Relevant and significant literature includes the study by Ong et al. [15], which used conjoint analysis to consider beer consumption in the Philippines. Their study showed that most people consuming alcoholic beverages are young respondents. To which, much of the discussion showed that the consumers’ casual enjoyment, celebration, and socialization are the reasons for alcoholic consumption in the Philippines. Their study suggested considering the generalization of alcoholic beverages rather than beer alone in the country using conjoint analysis. In accordance, Swahn et al. [16] discussed the income range of families in the Philippines, which influenced the younger generation to start consuming alcohol. The exposure of alcoholic beverage marketing to youth consumption was presented. The study by Pagkatipunan [12] also looked at the relationship between the younger generation and alcohol consumption, showing that most have consumed alcohol by the age of 13. It was suggested that a yearly identification of alcohol consumption should be monitored. On the other hand, Gregorio et al. [17] considered alcoholic beverages in the Philippines but focused on the baseline scientific information on volatile organic compounds. Amui and Etter [18] discussed the policies on tobacco and alcoholic beverages in the country’s health system. In other countries, it was seen that the behaviors of consumers shaped their consumption [19]. Gislason et al. [20] analyzed beer consumption in Denmark using conjoint analysis. Their study showed significant attributes to beer consumption preference among packaging and design. More attributes should be tackled and considered—especially in different countries due to their culture, influences, and taste recognition. 

According to Corduas et al. [21], attributes play a vital role in customers’ preferences when it comes to purchasing products. The study analyzed the attributes relevant to customers when buying wine and what affects their purchasing decisions. The attributes considered were food pairing, aroma/bouquet, wine complexity, alcoholic degree, color, etc. Applying the CUB model, the results indicated that grape variety and region of origin are relevant when purchasing wine products.

Given the various works and literature available, there is insufficient information regarding consumer behavior on their preference for alcoholic beverages. Although a study was conducted to understand customer preference in purchasing wine products, no studies identified attributes and factors that affect customer preference in purchasing alcoholic beverages such as wine, beer, and others.

Understanding customer preference is critical knowledge a company must attain. Strategies and methodologies were used to study customers’ behaviors toward a particular product. Moreover, analysts use a Conjoint Analysis Approach to study customer preference. This methodology is used in market research to identify how customers value the components or characteristics of a particular product or service [22]. According to Hauser and Rao [23], conjoint analysis allows analysts to transform holistic judgments into interval scales for every essential attribute based on customers’ preferences. For instance, Ong et al. [24] explored consumer preference for Milk Tea products in the Philippines. It revealed that as an attribute, the pearl size is considered the most by customers when purchasing milk tea products. Thus, applying conjoint analysis helps determine customer preference for alcoholic beverages and the relationship of their attributes to one another.

This study aimed to understand customer preference toward alcohol consumption using Conjoint Analysis. This study used different attributes that consumers consider when purchasing various alcoholic beverages. The attributes identified for this study are as follows: type of alcoholic beverage, oral sensation, flavors, origin, color, and price. In addition, the specific objectives of the study are described: (1) to determine the factors that affect customer preference when purchasing alcoholic products, (2) to evaluate the attributes and levels considered in alcohol consumption, (3) to apply conjoint analysis to determine the best combination of levels per attribute (stimuli) in alcoholic beverages based on customer preferences, and (4) to provide marketing strategies suitable using the results of this study.

The relevant information found in this study would be beneficial to the following: (1) companies that manufacture alcohol products as they will be given information regarding customer preference on alcohol attributes; (2) companies that sell alcohol products as this study identifies the most consumed alcohol brands preferred by younger consumers in the Philippines; (3) future researchers that will apply the same methodology as the results will highlight the most important combination of alcohol that the younger generation would consider in alcoholic beverage consumption.

## 2. Materials and Methods

### 2.1. Conceptual Framework

This study mainly comprises three (3) significant phases, as seen in Figure 1. The first phase reviews the possible attributes and levels considered in alcohol consumption based on the customer’s perspective. From this, possible combinations or alternatives of attributes called stimuli are generated by applying Conjoint Analysis. This information will be used in the online survey distributed by the researchers for its final phase. Due to the current circumstances, this study used purposive sampling techniques to gather data from respondents through online surveys. Additionally, according to Sethuraman et al. [25], an online survey is proven to be more effective in gathering data for a conjoint study.

Subsequently, the data underwent conjoint analysis, descriptive, and inferential statistics. Descriptive statistics are applied to understand the frequency of the responses, such as factors included in the demographics. On the other hand, inferential statistics, specifically the ANOVA test, was applied to determine factors that affect the differences in alcohol consumption. Furthermore, conjoint analysis was applied to identify the best attribute set for alcohol consumption.

### 2.2. Data Gathering

This study aimed to collect essential information from consumers, specifically customers that purchase alcohol products, through an online survey [24,25]. Due to the legal age implemented by the government in the Philippines, the minimum age of the respondents was at least 18 years old. Moreover, this study obtained at least 200 respondents that were evaluated through the stimuli generated by the SPSS Software. A minimum of 200 respondents may be used to generalize the conjoint analysis findings [24]. 

Collecting the data online, this study was conducted in the Philippines, focusing on young consumers. With Filipinos being this study’s target respondents, Sethuraman et al. [25] explained how collecting the data online would be more effective and sufficient for conjoint analysis. This study, therefore, utilized Google forms, distributed in different social media platforms and mobile applications such as Viber, Facebook, Twitter, Instagram, and Telegram. Data collection, which lasted for about a week, considered respondents aged 18 and above. The collection of data is not limited to a specific region. This study was conducted in the Philippines since the researcher would like to assess the behavior of Filipinos’ customer preferences towards alcohol consumption as it is one of the countries that mostly consume alcoholic beverages [14].

### 2.3. Statistical Treatment

Descriptive statistics were applied to create a visual presentation of the demographics of the respondents. The ANOVA test was also used to determine whether the demographics influence customers’ preferences towards alcohol consumption. Furthermore, the Correlation Test and Kendall’s Tau were determined using the SPSS 25 statistics software to test the relationship of the attributes and levels to alcohol preference and the reliability of the results, respectively.

### 2.4. Conjoint Design

The conjoint analysis was completed using the SPSS Statistics software, and statistical analyses such as conjoint, correlation, and Kendall’s tau were executed. Table 1 shows the attributes (alcohol types, oral sensation, flavors, alcohol origin, color, and price) and levels identified in this study for alcoholic beverages. The stimulus or set of combinations based on the attributes and levels generated by the software are presented in Table 2.

The study identified six (6) attributes customers consider when purchasing alcoholic beverages, gathered from various related studies and literature. The first attribute classified is the type of alcohol, an essential factor in alcohol consumption. According to Maldonado-Molina et al. [26], customers tend to select different types of alcoholic beverages, which causes a difference in consumption frequency. Numerous studies have been conducted to understand customers’ intentions in choosing various alcoholic drinks. In this study, the type of alcohol consumed was included so that the best combination for an alcoholic beverage can be determined to verify the type of alcohol that is the most preferred by consumers. The levels considered for this attribute are wine, beer, hard liquor, mixed drinks, and malt liquor.

The second attribute is oral sensation, which has always been one of the features customers consider when consuming alcoholic beverages. The oral sensation is the mouth’s reaction to the taste of the beverage being consumed by a person. According to Cravero et al. [27], different studies have focused on understanding the relationship between oral sensation and alcohol consumption. Although the studies were for medical reasons such as tongue-related studies and addiction, it is still evident that oral sensation influences the customers’ liking of certain alcoholic beverages. The study [20] also stated that oral sensation was found to be accountable for the differences in alcohol consumption. This study included three (3) levels: sweetness, bitterness, and astringency. Due to budget constraints, these three attributes are the most common taste attributes that people in the Philippines would consider, especially for younger generations. As indicated in the study by Ong et al. [15], younger generations would consider alcoholic beverages for social gatherings and celebrations and mainly prefer the alcoholic content and price. Consumers are less likely to be appealed by the flavor [12,16]. Regarding preference, most Filipinos would prefer a sweet taste and flavor when it comes to beverages [27], but this has not been established for alcoholic beverages since the components used in the of making alcohol do not have the sugar-like taste of sweetness and therefore may be different. In addition, the study by Canon et al. [28] presented how the phenolic reaction present in the fermentation of alcoholic beverages such as wine promotes different mouthfeels. To which, the phenolic compounds used in the brewing of alcoholic beverages would result in bitterness or astringency but also sweetness during the aging process of the alcoholic beverages [29]. The acidic reaction involved in the process of fermentation induces different mouthfeels. Tannins, for example, promote sweetness due to the hydrolysable tannins in wines. The oxidation and carbocation processes provide the astringency and bitterness of alcoholic beverages, and the cleavage of ring structures within the fermentation process also affects the mouthfeel [30].

For the third attribute, flavors have been the main driver for people to consume products such as alcoholic beverages. In recent studies, flavors were proven to influence the frequency of alcohol consumption by a consumer. Evidence has shown that it is an element for customers to consider when purchasing products [30]. Likewise, according to Sloat [31], flavors caused differences in profits made by alcohol companies. In her article, customers consume four typical flavors. There are Apple, Strawberry, Lime, and none. The study also used the same levels for the third attribute. As explained in the article by Yañez [32], the most popular alcoholic beverage is the San Miguel Corporation’s beer brand which is only about less than USD 1 or about PHP 40. Most malt drinks have been available with no flavor, but the San Miguel Corporation provided different flavors including Apple and Lime or Lemon as an innovation. Common drinks consumed by the young generation in the Philippines are another type of beer brand called Tanduay Ice and Smirnoff Mule, including other flavors such as strawberry. Since these are the top brands alongside Red Horse beer, San Miguel light beer (with no flavor), and brew kettle, to mention some, only the four flavors as levels were considered [33].

According to Corduas et al. [21], the origin of the alcohol product is considered one of the most influential attributes affecting alcohol consumption. Furthermore, expectations might vary depending on where the product originated, resulting in a difference in consumption. The study by Hollebeek et al. [34] arrived at the same conclusion stating that customers’ purchase intentions are also affected by the product’s origin. With this, the study considered the origin of the alcohol product as the fourth attribute but only considered two levels: local and foreign.

For the fifth attribute, the color of the alcoholic beverage was another factor influencing alcohol consumption. The color of the alcoholic beverage is the visual description of the liquid or alcohol consumed by customers. In the same study by Corduas et al. [21], color has been proven to be an objective factor influencing wine consumption. Thus, this paper would like to assess if the said attribute can affect the selection of alcoholic beverages. The levels considered in the study are as follows: colorless, light color, and dark color.

Lastly, the beverage price was identified as an attribute affecting alcohol consumption for the sixth attribute. A study by Xu and Chaloupka [35] explained that the prices of alcohol affect its demand and, thus, consumption. With this, the researchers included price as an attribute and considered three (3) levels based on Numbeo’s [36] estimation of alcohol prices in the Philippines. These levels are as follows: less than PHP 400, PHP 400–PHP 700, and more than PHP 700.

The attributes and levels mentioned above were encoded into the SPSS Statistics software and generated a set of combinations of orthogonal design, called stimulus, as shown in Table 2. The SPSS Software generated an optimal total of twenty-five stimuli and added two holdout cases. The holdout cases will be utilized to verify the accuracy of the results. As aforementioned, the stimulus will be evaluated by the respondents through an online survey using a 7-point Likert scale [24].

The collected information from the survey was encoded into the SPSS Software. With this, the identity of the importance score, utility score, Pearson’s R, and Kendall’s Tau would be considered. The importance score indicates the value of an attribute considered when a customer purchases an alcoholic beverage. The higher the importance score is, the more influential it is to alcohol preference. On the other hand, the utility score indicates how much value a customer perceives at a specific level per attribute. Similar to the importance score, the higher the utility score is, the more influential or essential it is. In addition, the researchers will determine Kendall’s Tau and Pearson’s R values. Kendall’s tau will indicate whether the results are reliable, while the correlation test will help the researchers determine the relationship between the observed and estimated preferences. Values greater than or equal to 0.8 for each test indicate strong and reliable results [37]. Moreover, through the stimulus ranking, the researchers can determine the best combination customers prefer as an alcoholic beverage.

### 2.5. Marketing Plan

Upon data gathering and conjoint analysis, the study provided a marketing strategy by applying Marketing Mix 5Ps. This tool helps determine and develop the best marketing tactics for a company. It suggests a company or product to consider which aspects can be changed or enhanced to better satisfy the target market’s needs and value and can be advantageous in the market competition [38].

The 5P’s in this tool stand for product, price, people, promotion, and place. The product is the object or service that clients are supplied with, and the price is the monetary value of the product. On the other hand, people refers to the company’s target customers, whereas promotion refers to the various tactics used to attract or present a product to a client. Finally, the place identifies the target market for which the corporation prefers to sell the goods [15]. All information included in the tool was based on the data gathered and conjoint analysis results.

## 3. Results

The study collected 205 respondents using the purposive sampling technique. As presented in Table 3, demographic profiling was constructed to summarize the respondents. Out of 357 responses collected, only 205 frequently (at least 4× a week) consume alcoholic beverages. Among the respondents, 53.66% (110) are female, while the remaining 46.34% (95) are male. For the age group, 74.63% of the respondents are between 21 to 25 years old. In addition, 104 out of 205 respondents are located in CALABARZON (Region IV-A), 69 out of 205 are from the National Capital Region (NCR), and the remaining are distributed in other regions listed in Table 3.

An initial assessment of the respondents’ preferences and alcohol consumption was summarized and shown in Table 4. Most of the respondents stated that they drink alcoholic beverages monthly or less. On the other hand, 58 out of 205 respondents drink alcoholic beverages 2 to 4 times a month. Furthermore, the survey also asked the respondents the number of alcohols they consume once they participate in drinking activities. Results indicate that 27.80% of the respondents drink ten or more glasses, 23.41% consume 1 or 2 glasses, and the remaining are shown in the table. Subsequently, the usual type of alcoholic beverage the respondents consume during drinking activities was also asked, in which 67 out of 205 respondents consume hard liquors, 61 prefer beers, 56 prefer mixed drinks, and the remaining are shown in the table below.

### 3.1. Conjoint Study

The study evaluated the data gathered and determined the respondents’ alcohol preferences based on importance and utility scores. Table 5 summarizes the importance score generated using the SPSS Statistics software. Results indicate that among the attributes included (e.g., Type of Alcoholic Beverages, Oral Sensation, & Flavors), the respondents mostly considered the product’s price. Thus, the product’s price positively influences consumers’ preference for alcoholic beverages. Seaman and Ikegwuono [39] stated that price controls how young adults drink alcohol, both in the amount consumed and the style in which the product is consumed. Their participation in drinking alcohol is highly influenced by their budget, affecting their preference to be more inclined toward the product’s price than the other attributes. Likewise, the study by Ong et al. [40] also mentioned a positive relationship between consumer buying behavior and prices. In this study, it is evident that the product’s price highly influences consumer preference, mainly since the demographics of the respondents are primarily categorized as young adults. 

Results also show that the type of alcoholic beverage comes second regarding its influence on customer preference. Although the product price has the highest importance score, as seen in Table 5, it is still evident that the difference in importance score between price and type of alcoholic beverage is minimal, with a value of 0.074. Many studies covered the impact of different alcoholic beverages on human behavior and health. Although there are instances wherein two different alcoholic beverages were compared, there is a lack of comparison between every type of alcoholic beverage. This study indicates customers prefer hard liquor over wine and beer, which is highly focused on related literature.

The third most important attribute based on the results of the SPSS is oral sensation. According to Sohani and Fahmy [41], the pandemic influenced how oral sensation influences a person’s liking of food products. It was stated in the article that under normal circumstances, factors such as price, taste, and brand were the primary basis of customer preferences. However, due to the pandemic and its influence on human health, many people have become conscious of what products they consume. As a result, oral sensation may still influence consumer preference, but it is not the primary basis of most products. The result of this study reflects the said concept in that even in alcoholic products, people tend to depend on factors other than oral sensation.

Following the attribute of oral sensation, the analysis indicates that the flavor of the alcoholic beverage ranks fourth in terms of its importance or influence on consumer preference. Similar to oral sensation, according to Running [42], a product’s flavor influences the consumption and preferences of a customer. In the same article, it was mentioned that oral sensation and flavor are correlated.

Furthermore, results indicate that the origin of the alcoholic beverage only influences consumer preference by 8.317%, which ranks sixth, as seen in Table 5. Consumer perceptions of product quality would be unaffected by country of origin, especially if other product attributes are highlighted during marketing and advertisement. Relating this concept to the study’s findings may explain why the product’s origin is only ranked sixth compared to the other attributes.

Lastly, results indicate that the product’s color has the lowest influence on consumer preference. Based on the results, the consumer only considers color by 7.790% in terms of its impact on their preferences toward alcoholic beverages. Many studies covered the influence of color on different product consumption and preferences. For instance, Carvalho et al. [43] stated that color significantly impacts the perceived consumer experience of beer. Likewise, Blackmore et al. [44] also stated that it provides information that influences consumer preference. Although a product’s color can affect consumer preference, this becomes the least preferred attribute as specific consumers depend on the perceived information related to other product attributes. Additionally, this explains why price and oral sensation are considered more compared to the product’s color. 

After distinguishing the importance level of each attribute, the study also determined the utility score, which establishes the vitality of a level under a particular attribute. Table 6 presents the summary of utility scores per attribute. The table shows the highest utility estimate and is considered the most preferred by the consumer.

Based on the generated SPSS software results, the most preferred type of alcoholic beverage is hard liquor, while the least preferred is beer. Both estimates of the initial assessment shown in Table 4 reveal that most respondents highly prefer hard liquor. Due to the effects of the pandemic on human health, many people have become conscious of what they intake. Based on the same table, beer is the least preferred alcoholic beverage. According to Dey et al., a low preference for beer products can be associated with risky drinking patterns [45].

Furthermore, results for the oral sensation attribute indicate that respondents highly prefer alcoholic beverages that provide a sweetness sensation. According to Ventura and Worobey [46], in most cases, people are biologically wired to like sweet tastes or sensations at an early stage of life. In comparison with sweetness, consumers prefer a product that would cause astringency once an alcoholic product is consumed.

For the flavor attribute, results indicate that consumers highly prefer strawberry while they prefer an alcoholic beverage with no flavor the least. Based on a study in the British Journal of *Addiction*, females prefer strawberry flavor the most, while men only prefer it next to apple-flavored beverages [31]. 

Subsequently, consumers prefer local brands to non-Filipino alcoholic brands. Local brands in low-involvement categories such as food are favored over worldwide brands since they cater to local tastes and requirements [47]. Likewise, Kalicharan [48] mentioned that locally manufactured products are highly preferred due to the patriotism and ethnocentrism of local citizens. 

Furthermore, results from the SPSS software show that consumers prefer alcoholic products with light colors while the dark-colored products are preferred the least. According to Carvalho et al. [43], people perceive a lower price on alcoholic beverages, specifically beers, if their visual appearances are pale or light in color. Thus, it explains the favorable findings toward light-colored drinks. Since the essential attribute is price, people tend to lean more toward light-colored beverages based on their perceived value of having a lower price. In addition, dark-colored beverages are identified to have a bitter taste compared to light-colored alcoholic products [44]. Thus, it supports the study’s findings that light-colored alcohol ranks higher than dark-colored alcohol as respondents prefers sweetness over bitterness. Therefore, this proves the perceived information that a product’s appearance affects their preferences, specifically since oral sensation is more prioritized than the product’s color.

Lastly, regarding the product’s price, consumers highly prefer less than PHP 400. Seaman and Ikegwuono [39] discussed in their paper that cheaper alcohol leads to greater consumption, especially during occasions. It can be highly observed in young adults, and relating it to the study’s demographics, most of the respondents are primarily young adults. It is evident that age positively influenced the preference for price levels since young adults tend to spend less money on alcoholic beverages.

Using IBM SPSS Statistics, the analysis also included Pearson’s R and Kendall’s Tau calculation. As mentioned, a value equal to or higher than 0.8 indicates a strong relationship between the variables. Based on the results of this study, both statistical tests had a value higher than 0.8, which means that the study results are reliable, and all variables had a strong relationship with one another. The results are presented in Table 7.

Upon identifying the utility estimate or score for each level in every attribute, the best combination in the stimulus was determined by simply substituting the utility score for the levels found in the combination. Table 8 shows the top five combinations with the highest score. Based on the study’s results, the best stimuli would be mixed drinks + Sweetness + Apple + Filipino + Light Colored + Less than PHP 400. With the results of the utility score, it is evident that the top five combination revolves around the top two highest utility estimates in each attribute. For instance, it can be observed that the type of alcoholic beverage found in Table 8 is either hard liquor or mixed drinks. Using Table 6 as a reference, it can be observed that these two levels are among the highest options for the type of alcoholic beverage.

### 3.2. Marketing 5P’s

After applying conjoint analysis and data gathering concerning the respondents’ demographics, a marketing plan was formed using the Marketing 5Ps. As previously mentioned, this tool included five distinct elements that would help the company or product strategize its success. As seen in Figure 2, the first element would be the product. Over the years, it has been proven that alcoholic beverages have increased consumption in countries such as the Philippines. 

The beverage alcohol market is very competitive, calling for the need to be strategic. Understanding consumer preference for alcoholic beverages is critical to success in this business. Results indicate that companies should provide products around the combination in Table 9. The results of the conjoint analysis indicate that most consumers tend to prefer hard liquor and mixed drinks over other types of alcoholic beverages. Despite this difference, it is undeniably a fact that companies should constantly monitor their product’s quality as this is the most important strategy observed in any given market [49].

## 4. Discussion

From the findings, the product’s price was evident among consumers who highly preferred alcoholic beverages under PHP 400. Although one combination included in Table 9 indicates that consumers also prefer a product that is between PHP 400–700, the conjoint analysis results showed that consumers would highly prefer a product that is below PHP 400. It is imperative to ensure that the product matches the price level since the importance score of the attribute, price, indicates it is the most influential factor that affects consumer preference. 

Following this element would be the place of the target respondents. Based on the data gathered, focusing on locations within region 4A (CALABARZON) and National Capital Region (NCR) is best. However, it is not recommended to limit the said areas since drinking an alcoholic beverage is a well-known activity in the Philippines. As mentioned, it is expected that in the coming years, the Philippines may highly encounter an increase in revenue based on alcoholic beverage sales Statista [14]. 

The next element included in the Marketing 5P’s would be product promotion. Recent studies suggest that social media platforms can be an excellent opportunity to advertise a product at a lesser cost, if not for free [50]. Likewise, free tasting should also be performed even for a limited sample size since it will not only help the company introduce the product to its clients, but it can also be an excellent opportunity to gain feedback that will surely help the company to adjust and match consumer preferences. According to Olenski [51], one of the most common strategies to become cost effective is to advertise products through online websites, better known as becoming an online company. Likewise, Lockett [52] investigated the impact of online marketing as a strategy for increasing sales through a cost-effective method of advertising products. Results indicated that online marketing provides a more convenient, viable, and influential method of building customer relationships, product promotion, and supplier relationships. With that, it is evident that companies should engage in online marketing to promote a product.

Finally, for the last element of the Marketing 5P’s, the target consumers would be Filipinos between the ages of 21 and 25 since most new marketing strategies in the Philippines, especially in the alcohol industry, tend to focus more on the youth due to their high prevalence of alcohol consumption [16]. In the same article, it mentioned that there is a significant relationship between the alcohol consumption of youths and alcohol marketing exposure. Findings indicate that due to the increase in alcohol use among young adults, alcohol companies shifted their focus to the said target consumers. With this, it would be necessary to focus on their preferences as it will help the company gain more revenue. Additionally, the data gathering results indicated that 74.63% of the respondents are within the said age group.

### 4.1. Practical Implications

The result of this study provided an in-depth insight into the current customer preferences of Filipino consumers towards alcoholic beverages. With the information found in this study, companies participating in the alcohol industry are given meaningful information on what consumers tend to focus on, specifically the product attributes. Companies can adjust the attributes of their products that will match consumer preferences. Conjoint analysis is an effective way of investigating the preferences of consumers towards a particular product using distinct attributes and corresponding levels. This study’s results provided the latest consumer preference trend toward alcohol.

### 4.2. Limitations and Future Research 

This study presented significant findings among the main young consumers of alcoholic beverages in the Philippines. Despite this, several limitations and improvements for study extensions may be considered. First, limited participants were considered due to the COVID-19 pandemic’s strict lockdown implementation. The authors would like to suggest that future studies utilize a larger sample size, using this study and its results as a benchmark for preliminary findings. In addition, it is recommended after the extension to determine whether the results would produce a different outcome than the findings found in this paper when strict face-to-face and social distancing health protocols are lifted. Moreover, the researchers recommend using different approaches or methodologies, such as applying Structural Equation Modeling (SEM) and K-Means Clustering for better market segmentation and behavioral aspects to be covered [53,54], especially since this study focused on sole preference analysis and marketing mix for strategy build up as a preliminary finding. Likewise, using different attributes and levels besides those included in this study is recommended. Since this study established the preferred combination from the type of alcohol, oral sensation, flavor, origin, color, and price, other studies may combine other attributes to highlight the best attributes to represent alcoholic beverages for generalization. This could be analyzed by conducting interviews and qualitative-quantitative mixed approaches for application and extension regarding beverages. Lastly, an extension using behavioral studies combined with the preference analysis may provide more insight and further suggestions for marketers. 

## 5. Conclusions

This study used six attributes to investigate consumer preference toward alcoholic beverages. These attributes were described as follows: Type of Alcoholic Beverage, Oral Sensation, Flavor, Origin, Color, and Price. Each attribute has its corresponding levels, as discussed earlier in this study. Results indicate that the product’s price influences customer preferences by 26.311% among the attributes mentioned. In terms of its levels, the conjoint analysis indicated that consumers highly prefer a product that is less than 400 pesos. This is followed by the type of alcoholic beverage having only a slight difference with the product’s price. This attribute influences consumer preference by 26.237%. For its levels, results indicated that Filipino consumers highly prefer hard liquor compared to the type of alcoholic beverages. Following the ranking of the attributes and levels are described: the third most influential would be the oral sensation, with 17.896%, and sweetness as the preferred level; the fourth influential attribute would be the flavor of the product, with 13.449% having strawberry as the preferred level; fifth influential attribute would be the origin of the product and Filipino-made as the preferred level; and lastly, the least important attribute would be the color of the product having light-colored alcoholic beverage as the preferred level. Upon identifying the importance and utility score of the attributes and levels, the study determined the best combination included in the Stimulus. Results indicate that the best combination would be the combined levels of mixed drinks, Sweetness, apple-flavor, Filipino-made, light-colored, and with a price of fewer than PHP 400. 

Furthermore, the conjoint analysis included a Pearson Correlation Test and Kendall’s Tau to determine whether there is a significant relationship among the variables and if the results are reliable. Results indicated that both tests had a value greater than 0.8, meaning there is a significant relationship among the variables and that the conjoint analysis results are reliable.

## Figures and Tables

**Figure 1 foods-12-01401-f001:**
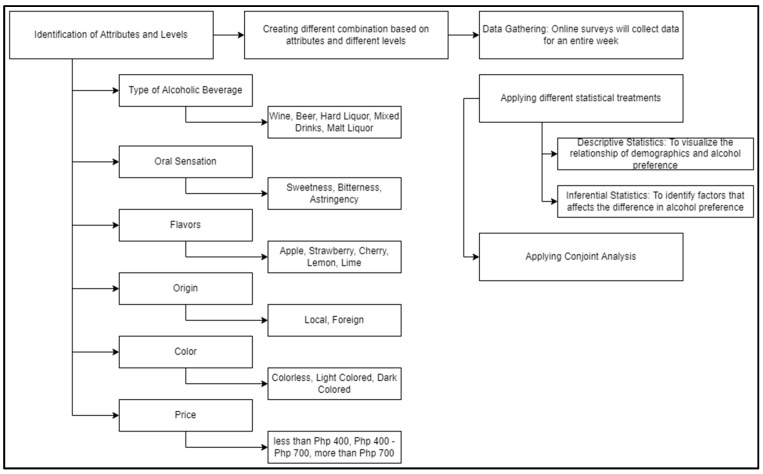
Conceptual Framework.

**Figure 2 foods-12-01401-f002:**
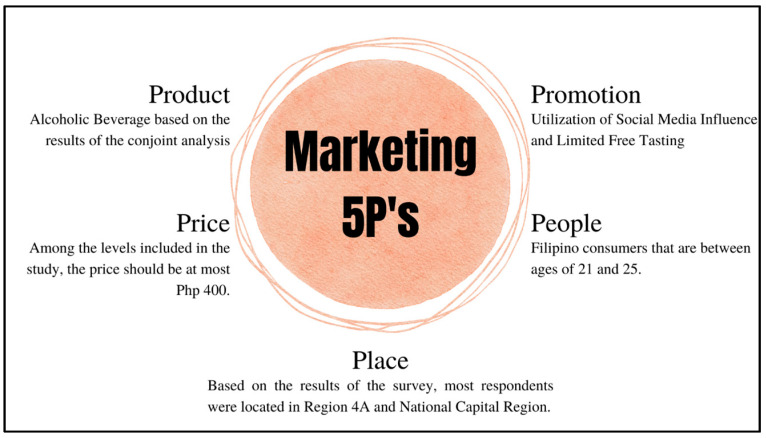
Marketing 5P’s.

**Table 1 foods-12-01401-t001:** Attributes and Levels for Alcohol Beverages.

Attributes	Levels
Type of Alcohol	Wine, Beer, Hard Liquor, Mixed Drinks, Malt Liquor
Oral Sensation	Sweetness, Bitterness, Astringency
Flavors	Apple, Strawberry, Lime, None
Origin of Alcohol	Local, Foreign
Color	Colorless, Light Color, Dark Color
Price	Less than Php 400, Php 400–Php 700, More than Php 700

**Table 2 foods-12-01401-t002:** Stimulus.

Combination	Type of Alcohol	Oral Sensation	Flavors	Origin	Color	Price
1	Mixed Drinks	Astringency	Apple	Non-Filipino	Colorless	Php 400–Php 700
2	Wine	Bitterness	None	Non-Filipino	Dark Colored	Less than Php 400
3	Malt Liquor	Sweetness	Strawberry	Filipino	Colorless	Less than Php 400
4	Mixed Drinks	Sweetness	None	Non-Filipino	Colorless	More than Php 400
5	Malt Liquor	Astringency	None	Filipino	Light Colored	Less than Php 400
6	Malt Liquor	Bitterness	Apple	Non-Filipino	Colorless	Php 400–Php 700
7	Malt Liquor	Bitterness	Apple	Filipino	Light Colored	More than Php 400
8	Mixed Drinks	Bitterness	Lime	Filipino	Light Colored	Php 400–Php 700
9	Hard Liquor	Bitterness	Apple	Non-Filipino	Colorless	Less than Php 400
10	Mixed Drinks	Bitterness	Strawberry	Filipino	Dark Colored	Less than Php 400
11	Hard Liquor	Bitterness	None	Filipino	Light Colored	Php 400–Php 700
12	Wine	Bitterness	Lime	Filipino	Colorless	More than Php 400
13	Beer	Astringency	Apple	Filipino	Dark Colored	More than Php 400
14	Hard Liquor	Astringency	Lime	Filipino	Colorless	Less than Php 400
15	Wine	Astringency	Strawberry	Non-Filipino	Light Colored	Php 400–Php 700
16	Beer	Bitterness	Strawberry	Filipino	Colorless	Php 400–Php 700
17	Wine	Sweetness	Apple	Filipino	Colorless	Less than Php 400
18	Beer	Sweetness	None	Filipino	Colorless	Php 400–Php 700
19	Wine	Sweetness	Apple	Filipino	Light Colored	Php 400–Php 700
20	Beer	Bitterness	Apple	Non-Filipino	Light Colored	Less than Php 400
21	Hard Liquor	Sweetness	Apple	Filipino	Dark Colored	Php 400–Php 700
22	Mixed Drinks	Sweetness	Apple	Filipino	Light Colored	Less than Php 400
23	Hard Liquor	Sweetness	Strawberry	Non-Filipino	Light Colored	More than Php 400
24	Malt Liquor	Sweetness	Lime	Non-Filipino	Dark Colored	Php 400–Php 700
25	Beer	Sweetness	Lime	Non-Filipino	Light Colored	Less than Php 400
26	Mixed Drinks	Astringency	Apple	Filipino	Light Colored	Php 400–Php 700
27	Hard Liquor	Sweetness	Lime	Filipino	Colorless	Less than Php 400

**Table 3 foods-12-01401-t003:** Demographic Profiling.

Demographics Factor	Category	N	%
Gender	Male	95	46.34
	Female	110	53.66
Total		205	
Age	18–20	24	11.71
	21–25	153	74.63
	26–29	19	9.27
	30–34	4	1.95
	35 or above	5	2.44
Total		205	
Location	National Capital Region (NCR)	69	33.66
	Ilocos Region (Region I)	2	0.98
	Central Luzon (Region III)	9	4.39
	Calabarzon (Region IV-A)	104	50.73
	Bicol Region (Region V)	2	0.98
	Western Visayas (Region VI)	4	1.95
	Central Visayas (Region VII)	12	5.85
	Eastern Visayas (Region VIII)	1	0.49
	Northern Mindanao (Region X)	2	0.98
Total		205	

**Table 4 foods-12-01401-t004:** Customer Preference & Alcohol Consumption.

Factors	Category	N	%
Frequency of Drinking Alcohol	Four or more times a week	7	3.41
	2 to 3 times a week	25	12.20
	2 to 4 times a month	58	28.29
	Monthly or less	115	56.10
Number of Alcohol Drinks in a Typical Day	Ten or more glasses a day	57	27.80
	7 to 9 glasses a day	35	17.07
	5 or 6 glasses a day	29	14.15
	3 or 4 glasses a day	36	17.56
	1 or 2 glasses a day	48	23.41
Usual Type of Alcohol Beverage Consumed	Wine (e.g., Carlo Rossi)	19	9.27
	Beer (e.g., Red Horse Beer)	61	29.76
	Hard Liquor (e.g., Black Label)	67	32.68
	Mixed Drinks	56	27.32
	Malt Liquor (e.g., Colt 45)	2	0.98

**Table 5 foods-12-01401-t005:** Summary of Importance Score.

Attribute	Importance Value	Ranking
Type of Alcoholic Beverage	26.237	2
Oral Sensation	17.896	3
Flavors	13.449	4
Origin	8.317	5
Color	7.790	6
Price	26.311	1

**Table 6 foods-12-01401-t006:** Summary of Utility Score.

Attribute	Level	Utility Estimate	Ranking
Type of Alcoholic Beverage	Wine	−0.061	3
	Beer	−0.177	5
	**Hard Liquor**	**0.201**	**1**
	Mixed Drinks	0.178	2
	Malt Liquor	−0.141	4
Oral Sensation	**Sweetness**	**0.144**	**1**
	Bitterness	−0.113	3
	Astringency	−0.031	2
Flavors	Apple	0.016	3
	**Strawberry**	**0.077**	**1**
	Lime	0.024	2
	None	−0.117	4
Origin	**Filipino**	**0.060**	**1**
	Non-Filipino	−0.060	2
Color	Colorless	−0.016	2
	**Light Colored**	**0.064**	**1**
	Dark Colored	−0.048	3
Price	**Less than PHP 400**	**0.190**	**1**
	PHP 400–PHP 700	−0.001	2
	More than PHP 700	−0.189	3

Note: Bold figures indicated highest attribute.

**Table 7 foods-12-01401-t007:** Summary of Statistical Results.

Statistical Test	Value	Significance
Pearson’s R	0.955	0.000
Kendall’s Tau	0.876	0.000
Kendall’s Tau for Holdout	1.000	

**Table 8 foods-12-01401-t008:** Best Combination.

Combination #	Score	Ranking	Combination
22	0.652	1	Mixed Drinks + Sweetness + Apple + Filipino + Light Colored + Less than PHP 400
27	0.603	2	Hard Liquor + Sweetness + Lime + Filipino + Colorless + Less than PHP 400
14	0.428	3	Hard Liquor + Astringency + Lime + Filipino + Colorless + Less than PHP 400
21	0.372	4	Hard Liquor + Sweetness + Apple + Filipino + Dark Colored + PHP 400–PHP 700
10	0.344	5	Mixed Drinks + Bitterness + Strawberry + Filipino + Dark Colored + Less than PHP 400

**Table 9 foods-12-01401-t009:** Marketing 5P’s product.

Combination #	Combination
22	Mixed Drinks + Sweetness + Apple + Filipino + Light Colored + Less than Php 400
27	Hard Liquor + Sweetness + Lime + Filipino + Colorless + Less than Php 400
14	Hard Liquor + Astringency + Lime + Filipino + Colorless + Less than Php 400
21	Hard Liquor + Sweetness + Apple + Filipino + Dark Colored + Php 400–Php 700
10	Mixed Drinks + Bitterness + Strawberry + Filipino + Dark Colored + Less than Php 400

## Data Availability

The data presented in this study are available on request from the corresponding author.

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
