# Peer review of "Application of Conjoint Analysis and 5Ps Marketing Mix in Identifying Customer Preference of Alcohol Consumption among Youths in the Philippines"

_foods, 2023, doi:10.3390/foods12071401_

Round 1

Reviewer 1 Report

The study is focused on a population between the ages of 18 and 25, so the results cannot be generalized to the Philippine population, without knowing its age distribution.

The frequency of alcohol consumption is not associated with age and may be an important parameter.

I don't see the point of adding a marketing mix proposal to market research, based on a segmented study.

The bibliographical references can be expanded and updated.

The authors do not include a Discussion section, in order to compare the findings with the results of other studies.

Author Response

Thank you for providing your constructive comments and suggestions. The article was revised extensively and is believed to be ready for publication. All comments were addressed as presented and highlighted in the revised manuscript. We highly appreciated the input given.

Reviewer 2 Report

The paper submitted reports a study that evaluates consumer preferences of alcohol beverages in the Philippines using conjoint analysis.
References cited in the introduction focus on historical and general aspects of alcohol consumption, aimed at providing a historical framework to the reader.
This approach doen not appear exactly appropriate for an audience of researchers, as it limits the description of the technical and methodological aspects that led the authors to choose the research methodology described in the paper.

The proposed methodology is consistent with other works reported in the references cited, and the results obtained, in the opinion of this reviewer, appear consistent with the materials and methods chosen. However, the claims of the authors intending to generalize the results obtained outside the collective analyzed (Philippine consumers) [see abstract lines 22-23 and lines 151-152] seem to overestimate the impact of the survey, both for the collective size and for the methodology followed (online questionnaire). Furthermore, the authors do not indicate what medium was used to publicize the survey.

In the conclusion paragraph, moreover, the authors recommend themselves potential methodological improvements, when it would have been more appropriate to state that the study presented should be considered preliminary to future research.

Furthermore, the paper is written in a very poor English form (even in the abstract), which makes it difficult to read and does not allow proper understanding of the text; thus, the paper requires an in-depth revision and rewriting with a thorough critical reading by a native English speaker.

Author Response

(The authors gave the same response as above.)

Reviewer 3 Report

The objective of this paper is to evaluate the consumer preference for alcoholic beverages in the Philippines. The introduction should be extensively edited as it mainly discusses the health risks of alcoholic beverages and does not review a large amount of research on consumers’ perceptions of alcoholic beverages. The methodology only included eight oral sensations (two are basic tastes) and five flavours in the conjoint analysis, which is a large limitation of this study. There are many flavours, tastes and sensations in alcoholic beverages and only to include eight clearly limits the scope of this work. In addition, the paper should be extensively edited for grammar. Overall, the paper has many limitations in the attributes and levels of the conjoint analysis task that needs to be addressed by the authors. Minor comments are listed below.

Line 27- “Alcoholic beverage” not alcohol beverage.

Line 43- “Only” seems unnecessary in the sentence.

Line 49- “Moreover” is used repeatedly throughout the introduction.

Line 37-98- These paragraphs do not seem to fit with the theme/objective of the paper.

Line 117 -121- There are many studies on consumers’ alcoholic beverage preferences that should be included in the introduction (including those that used conjoint analysis).

Line 181- Data collection lasted.., instead of “will only…”

Line 221- Sweetness and bitterness are basic tastes. Astringency is the only sensation listed. Also, there are many other tastes, flavours and sensations included in alcoholic beverages. The authors should explain why only these 3 were included.

Line 227- As stated above, there are many different flavours, how were these 5 chosen over the many other flavours present in alcoholic beverages?

Line 250- Usually the attributes and levels are chosen based on the literature review, but the authors included only a few studies to select their attributes and levels. Why were more studies not included?

Line 295- Were the participants screened for alcohol consumption? What if they did not usually consume alcohol?

Line 493- Any limitations to the study?

Author Response

(The authors gave the same response as above.)

Round 2

Reviewer 1 Report

The title, the abstract and the paper continue to cite the population of the Philippines, taking into account age ranges that include 4 people (30-34 years) or 5 people (population over 35 years), which surely means more than 50% of the population.

I do not consider the development of a marketing plan as part of a research paper. It is a consulting proposal.

Author Response

Thank you for your highlights and comments. The revisions were made and highlighted in the revised manuscript.

Reviewer 3 Report

Thank you to the authors for addressing my comments. I still have a couple of comments:

The paper should be edited for grammar/spelling.

Line 231- So why were bitterness and astringency included? The reference said the consumers consider sweet taste and flavour.

Line 234- Needs to be more explanation, how can the flavours of the whole alcoholic beverage category be covered by four flavours (apple, strawberry, lime, none)? As stated in my previous review this is the main limitation of this work and an explanation needs to be included in the text.

Author Response

(The authors gave the same response as above.)
